# Zebrafish: A Model to Study and Understand the Diabetic Nephropathy and Other Microvascular Complications of Type 2 Diabetes Mellitus

**DOI:** 10.3390/vetsci9070312

**Published:** 2022-06-22

**Authors:** Charles Sharchil, Amulya Vijay, Vinu Ramachandran, Sambhavi Bhagavatheeswaran, Reena Devarajan, Bhupendra Koul, Dhananjay Yadav, Anandan Balakrishnan

**Affiliations:** 1Department of Genetics, Dr. ALM PG Institute of Basic Medical Sciences, University of Madras, Chennai 600113, Tamil Nadu, India; andrewchales@gmail.com (C.S.); amulyav16@gmail.com (A.V.); vinutwin@gmail.com (V.R.); sambhavi.sam@gmail.com (S.B.); 2Department of Clinics, Madras Veterinary College, Tamil Nadu Veterinary and Animal Sciences University, Vepery, Chennai 600007, Tamil Nadu, India; reenadevarajan2006@yahoo.co.in; 3School of Bioengineering and Biosciences, Lovely Professional University, Phagwara 144411, Punjab, India; bhupendra.18673@lpu.co.in; 4Department of Medical Biotechnology, Yeungnam University, Gyeongsan 38541, Korea

**Keywords:** diabetes mellitus, microvascular complications, diabetic nephropathy, zebrafish, animal model

## Abstract

**Simple Summary:**

Diabetes is a chronic metabolic disease characterized by high blood glucose levels (hyperglycemia). Type 2 diabetes mellitus (T2DM) and its complications are a worldwide public health problem, affecting people from all developed and developing countries. Hyperglycemia can cause damage to the vascular system and dysfunction of organs, such as the kidneys, heart, retina of the eyes, and nerves. Diabetic nephropathy (DN) is one of the most severe micro-vascular complications, which can lead to ESRD (end-stage renal disease). Zebrafish are ideal for wide-scale analysis or screening, due to their small size, quick growth, transparent embryos, vast number of offspring, and gene similarity with humans, which combine to make zebrafish an ideal model for diabetes. The readily available tools for gene editing using morpholinos or CRISPR/Cas9, as well as chemical/drug therapy by microinjection or skin absorption, enable zebrafish diabetes mellitus models to be established in a number of ways. In this review, we emphasize the physiological and pathological processes relating to micro-vascular problems in zebrafish, as well as the many experimental zebrafish models used to research DN, and the DN-related outcomes and mechanisms observed in zebrafish. This study specifies the benefits and drawbacks and future perspective of using zebrafish as a disease model.

**Abstract:**

Diabetes mellitus (DM) is a complicated metabolic illness that has had a worldwide impact and placed an unsustainable load on both developed and developing countries’ health care systems. According to the International Diabetes Federation, roughly 537 million individuals had diabetes in 2021, with type 2 diabetes mellitus accounting for the majority of cases (T2DM). T2DM is a chronic illness defined by insufficient insulin production from pancreatic islet cells. T2DM generates various micro and macrovascular problems, with diabetic nephropathy (DN) being one of the most serious microvascular consequences, and which can lead to end-stage renal disease. The zebrafish (*Danio rerio*) has set the way for its future as a disease model organism. As numerous essential developmental processes, such as glucose metabolism and reactive metabolite production pathways, have been identified in zebrafish that are comparable to those seen in humans, it is a good model for studying diabetes and its consequences. It also has many benefits over other vertebrate models, including the permeability of its embryos to small compounds, disease-driven therapeutic target selection, in vivo validation, and deconstruction of biological networks. The organism can also be utilized to investigate and understand the genetic abnormalities linked to the onset of diabetes problems. Zebrafish may be used to examine and visualize the growth, morphology, and function of organs under normal physiological and diabetic settings. The zebrafish has become one of the most useful models for studying DN, especially when combined with genetic alterations and/or mutant or transgenic fish lines. The significant advancements of CRISPR and next-generation sequencing technology for disease modelling in zebrafish, as well as developments in molecular and nano technologies, have advanced the understanding of the molecular mechanisms of several human diseases, including DN. In this review, we emphasize the physiological and pathological processes relating to microvascular problems in zebrafish, as well as the many experimental zebrafish models used to research DN, and the DN-related outcomes and mechanisms observed in zebrafish.

## 1. Introduction

Diabetes mellitus (DM) is a multifactorial disease condition that affects a large population. Among the diabetic patients worldwide, the majority, approximately 90–95%, of cases are Type 2 diabetes mellitus (T2DM). The World Health Organization (WHO) global reports state that the number of adults having diabetes has increased almost four times since 1980, to 422 million, and it will increase to 693 million in another two decades [1]. The disease is characterized by high blood sugar levels, due to dysregulated insulin activity, which makes the managing of glycemic levels difficult. As per reports and research, though the key problem of T2DM is characterized as reduced β-cell function, long time non-insulin-dependent diabetes is characterized by insulin resistance and β-cell dysfunction [2].

DM mainly leads to vascular complications involving both the macrovascular, which involves arteries and veins which are large vessels, as well as the microvascular, (involving small vessels, such as capillaries) networks. It is the high glucose level or the hyperglycemic levels that play a significant role in initiating vascular complications in diabetic patients, through many metabolic and hemodynamic pathways [3]. These vascular complications are one of the significant causes leading to morbidity and mortality in individuals with DM. In addition, various other long-term complications apart from nephropathy, such as cardiovascular complications, blindness, and kidney failure have been associated with DM patients, and which decrease their overall quality of health and life [4].

Diabetic nephropathy (DN) is a condition leading to chronic kidney disease in diabetic patients. Due to hyperglycemia, reduced renal function and increased albumin excretion ratio arise in diabetic kidney disease (DKD). The connection between diagnosing diabetes and the onset of kidney disease can help in distinguishing between diabetic-specific and non-specific DKD, based on the increased levels of albumin and decreased levels of the estimated glomerular filtration rate (eGFR), which are two important clinical markers for checking kidney function [5]. The kidneys tend to encounter various modifications, including deposition of extracellular matrix, thickening of the glomerular basement membrane, changes in proliferation followed by tubular atrophy, and finally ending in interstitial fibrosis and glomerulosclerosis, leading to scarring in the kidneys [6]. The development of DKD involves a hemodynamic pathway, involved in the glomerular hemodynamic changes that occur in DN [7]. Metabolic pathways are also promoted within the diabetic kidney and activate various molecules such as cytokines, growth factors, advanced glycation end products, angiotensin-II, monocyte chemoattractant protein, and reactive oxygen species [8]. These pathways ultimately lead to renal histological changes in the glomeruli of DN patients.

A worldwide upsurge of T2DM over the next few decades will significantly increase the occurrence of DKD, with the global prevalence of microalbuminuria in diabetes already being 39%. DKD is reported to be developing in about one-third of people who are affected with T2DM. Certain ethnic groups, such as Native Americans, Indigenous Australians, and African Americans living in developed countries have a greater prevalence for end-stage renal disease (ESRD) due to diabetes than in developing or underdeveloped countries [9]. With the increase in the prevalence of DKD in diabetic patients, it is the need of the hour to look into the exact mechanism leading to the condition and also to discover possible strategies and therapeutics to halt the development of this condition. Researchers worldwide are creating various experimental setups and experimental models, including the use of animal models, to study the diabetic nephropathy condition. Various animal models are used to study the condition, and one such animal model that has gained recognition over the past few decades is the zebrafish.

The zebrafish (*Danio rerio*), which is a small tropical freshwater fish, has gained its place in the field of research due to its ease of maintenance and various other factors that makes it an extensively used animal model in biomedical research. The embryos are transparent, and the development of the organs is rapid as seen in common vertebrates and visible within 48 h post-fertilization (hpf) [10]. These are some of the key useful properties seen in zebrafish that makes it an ideal model to carry out manipulation studies and in-situ observation. In comparison with the human kidney, the zebrafish embryonic kidney possesses pronephros (Figure 1) that are in a well-developed state and become functional by 48 hpf, accomplishing the blood filtration and osmoregulation functions at the larval stages [11].

Structurally, the blood filtering glomerulus, the two proximal and distal tubules, and the pronephrotic duct present in the zebrafish are similar to that of the mammalian nephron. Zebrafish have a segmented tubular epithelium that exhibits ducts and transporters in a manner that differentiates each segment with important physiological functions and secretory properties to each nephron segment similarly to metanephric nephrons [12]. The zebrafish pronephros consists of parts that are similar to that of the mammalian nephrons, which include the neck, proximal convoluted tubule (PCT), proximal straight tubule (PST), distal early, distal late, corpuscle of Stannius, and the pronephric duct, with the exclusion of the corpuscle of Stannius [13]. Considering the similarity of zebrafish with humans, this model organism is used for studying developmental and functional biology, as well as for drug screening, rejuvenation, and disease modeling studies.

Animal models, particularly zebrafish (*Danio rerio*), are critical for study, since they represent the majority of the anatomical, functional, and biochemical aspects of human disease. For a new study design, the selecting and understanding of an animal model is essential. When evaluating novel chemicals for their medicinal potential, screening models are critical. The optimal screening model has identifiable human-related traits and mimics the etiology of disease. Several significant biomarkers implicated in the development of diabetes complications have recently been found using in vitro and in vivo zebrafish models. Diabetic microvascular problems such as diabetic retinopathy, diabetic neuropathy, and diabetic nephropathy are well studied using zebrafish. Each animal model has its own set of benefits and drawbacks. The current paper describes the use of zebrafish as a model for evaluating microvascular complications in diabetes.

## 2. Studying Diabetes Using Zebrafish Models

Long-term T2DM is caused by insulin resistance and β-cell dysfunction, followed by β-cell loss. T2DM models of zebrafish have been created using a variety of methods, including diet and genetic approaches. Owing to its simplicity, the approach of immersing zebrafish in a solution containing glucose has been proven to be an easy procedure [14]. The exocrine and endocrine portions of the zebrafish pancreas are similar to those of mammals, and it is capable of regeneration; therefore, it has been utilized to research the pathophysiology of T2DM [15]. α-cells, β-cells, δ-cells, ε-cells, and pancreatic polypeptide cells make up the endocrine compartment. The development, signaling pathways, and processes of the endocrine pancreas in zebrafish are quite similar to those in mammals (Figure 2) [16]. The zebrafish’s conserved pancreatic anatomy and glucose management system have led to the discovery of new diabetes targets.

The pancreatectomy approach is a rare technique that was used to develop a transgenic zebrafish model with a green fluorescent protein (GFP) selective islet expression. Intraperitoneal injection of streptozotocin (STZ), which may successfully ablate cells at greater concentrations, is a typical means of inducing chemical-induced diabetes in rodents; while, apart from rodents, STZ induction in a zebrafish model was also carried out by Intine et al., although due to various limitations the procedure could not be reproduced later [35]. Alloxan is yet another substance that can destroy cells in both adults and larvae of zebrafish [36].

Insulin promoter is known to induce apoptosis by activating a doxycycline/ecdysone-dependent transcription factor, as well as the TetOR-based promoter. One strategy for creating insulin-resistant transgenic mice is to express a dominant-negative *IGF-I receptor* (IGFIR) in skeletal muscle, which results in an elevated fasting glucose level (Transgenic line (Tg) (acta1:dnIGF1R-EGFP)). Another method for creating transgenics is to use the CRISPR/Cas9 approach to knock down the liver-specific insulin receptor Tg (actb2:cas9; (*U6x*:*sgRNA*(insra/b), resulting in hyperglycemia after postprandial, and hypoglycemia after fasting [18].

GFP-expressing zebrafish strains are widely utilized to research pancreatic development and glucose homeostasis [16,37]. GFP expression in the zebrafish preproinsulin promoter and preproglucagon promoter is a marker for cells and pancreatic cells [38]. It is simple to evaluate changes in β-cell and α-cell mass and number with these cell-specific transgenic lines [39,40].

In general, zebrafish have contributed to a better understanding of the major etiological processes of diabetes, to develop therapeutic approaches. Zebrafish have a lot of potential advantages, such as their regenerative capacity and customized medical applications, making them a promising animal model in this field [41,42].

### Glomerular Development and Disease Model

To maintain normal renal function, the glomerular basement membrane (GBM) creates cell-to-cell connections in the presence of glomerular capillaries, which constitute the slit diaphragm filtration barrier. Glomerular dysfunction is caused by damage to this area. Through podocyte destruction and structural alterations, damage to podocytes lowers the filtration barrier function. Proteinuria is linked in patients with focal segmental glomerulosclerosis, minimal change disease, and diabetes [43].

End-stage renal disease (ESRD) is caused by glomerular degeneration in humans. Over the last two decades, genetic studies of human diseases with glomerular injury and various knockout animal models have discovered numerous genes (Table 1) crucial for kidney formation and function [44].

The relevance of podocytes in the zebrafish pronephros was investigated by Kramer-Zucker et al. [19]. Morpholino-mediated *nephrin* and podocin knockdown resulted in podocyte foot process effacement, slit diaphragm loss, and glomerular filtration barrier permselectivity loss. The zebrafish pronephros has been used as a classical system for research on podocyte maturation and glomerular filtration barrier integrity.

Crb2b is linked to the crumb protein family, which is involved in the development and maintenance of apical-basal polarity in zebrafish epithelial cells [45,46]. The podocyte structure in zebrafish is dramatically disrupted by morpholino-mediated suppression of the Crb2b protein, the slit diaphragm is lost, and the crucial slit diaphragm protein nephrin is misoriented to the apical projections inside the morphant podocyte. As a result, apical-basal podocyte polarity modulation and establishment are functionally significant and must be explored as a reason for glomerular dysfunction [19].

Cilia are required for normal organ growth and function. In response to kidney injury, the length of primary cilia in mammalian renal tubular cells raises dramatically during acute tubular necrosis and ureteral obstruction [47,48,49]. *Foxj1a* is a master regulator of genes involved in the creation and function of motile cilia [50]. The zebrafish gene *foxj1a* is necessary for cilia motility. *Foxj1a* morpholino knockdowns were performed on WT strain TU-AB zebrafish embryos at one- to four-cell stage. The cilia motility target genes efhc1, tektin-1, and dnahc9 were stimulated by overexpression of *foxj1a* in blocked tubules. Loss of *foxj1a* embryos resulted in a failure to overexpress efhc1, tektin-1, and dnahc9, as well as a loss of improved cilia beat rates following obstruction, indicating that foxj1 plays an important role in cilia function after injury. Multiple types of tissue damage are caused by a Foxj1 transcriptional network of ciliogenic genes, according to research [20].

## 3. Modelling Diabetic Nephropathy Using Zebrafish

By exchanging guanine nucleotides for GTPase Rac1, the ELMO1/DOCK180 complex aids in cell migration regulation [51]. ELMO1 is involved in the early stages of vascular formation [52]. In vitro and in vivo studies have revealed that the ELMO1 gene activates the Rac1/PAK/AKT signaling cascade, to protect endothelial cells from apoptosis [53]. To explore renal phenotype, CRISPR/Cas9 induced ELMO1 deletion was performed in 48 hpf embryos. The pronephric structure was altered in hyperglycemic circumstances and ELMO1 knockout embryos, impacting renal system function. ELMO1 protects the glomerulus from apoptosis and damage caused by hyperglycemia. Meanwhile, ELMO1 overexpression has been associated with extracellular matrix protein buildup in diabetic patients, resulting in diabetic nephropathy [21].

Insulin production was measured in overfed Tg(1.0ins: EGFP)sc1 zebrafish by assessing the EGFP signal strength, and IP and OGT tests revealed increased FBG with glucose intolerance and insulin resistance. In DIO zebrafish, metformin and glimepiride reduced hyperglycemia. Rapid onset of T2DM, favorable responsiveness to anti-diabetic therapy, and pathogenic resemblance in transcriptome pathways to the human platform are all advantages of the T2DM zebrafish model. As a result, it is possible to employ zebrafish as a model for human T2DM [54].

### 3.1. Molecular Level Approaches to Model DN in Zebrafish

#### 3.1.1. Genes

In zebrafish, *Pdx1* regulates embryonic glucose metabolism by activating insulin gene expression in developing pancreatic progenitor cells; it was also demonstrated, in a recent study conducted by Wiggenhauser et al., that *pdx1* Knockout could lead to a phenotype similar to that of DN in Zebrafish, and they also identified phosphatidylethanolamine as a metabolite that could potentially be promoting early diabetic kidney damage [55]. Adult *pdx1* mutant zebrafish have smaller bodies, lower survival rates, and lower amounts of beta cells and insulin. The *pdx1* mutant beta cells were nutrient-sensitive and experienced enhanced apoptosis when they were fed a high-fat diet. This model will let researchers test new therapy strategies and continue to learn more about the molecular pathways that cause hyperglycemia’s harmful effects.

Compound transgenics were created using the *pdx1sa280* mutant allele in conjunction with NeuroD:eGFP immunofluorescence labelling. In mutants, insulin expression and islet size were reduced. *Pdx1* mutants were tested for glucagon-producing α cells at 72 h, because these play a function in islet cell fate allocation [56], and loss of *Pdx1* has been linked to cell identity alterations [57]. The number of cells in mutants was reduced, but not the ratio of α to β cells [58]. Early endocrine cell specification was generally intact in *pdx1* mutants in an previously revealed report on *pdx1* knockdown [22].

Similar to humans and mice, the glomerulus of zebrafish contains endothelium, mesangial cells, and podocyte cell types [59], as well as proteins, including *nephrin* and podocin [60]. Recent research [61] indicated that *zApoL1* is expressed in podocytes and, to a lesser degree, in endothelial cells in human biopsies, along with in the glomerulus of zebrafish larvae. When zebrafish larvae had their *zApoL1* expression knocked down, they developed severe pericardial edema, indicating impaired kidney function. It was also shown that a glomerular phenotype can be caused by an absence of APOL1. Podocytes are notably impacted in ESRD; it is highly likely that the *zApoL1* of endotheliumplays a modest role in the development of glomerulopathies at some point [23].

*SRGAP2a* is a major “hub” gene that is strongly related to proteinuria and eGFR in DN patients [62,63]. It is a crucial component of the Slit/Robo signaling pathway during neural development and is largely found in podocytes. *SRGAP2a* and SRGAP2b are the two SRGAP2 orthologs found in zebrafish; however, only *SRGAP2a* has substantial homology with human *SRGAP2a* [64]. p53 MO was co-injected with *SRGAP2a* splicing MO targeting exon 1 and intron 1, to minimize the cell mortality caused by off-target effects of MO (e1i1). At 3.5 days after fertilization, zebrafish larvae had 85.3 percent total pericardial, periorbital, and body edema (dpf). *SRGAP2a* knockdown caused podocyte foot process effacement, a damaged slit diaphragm, and a disordered glomerular filtration barrier in transgenic embryos Tg (pod:GFP). The confocal imaging revealed that GFP expression was reduced. Injection of *SRGAP2a* mRNA was shown to reverse the MO-induced phenotype and decrease overall edema. The deletion of *SRGAP2a* led to zebrafish podocyte foot process effacement and disruption of the glomerular filtration barrier (GFB) [24].

Several enzyme systems, including aldehyde dehydrogenase (ALDH), Aldo-keto reductase (AKR), and glutathione s-transferase (GST), are responsible for acrolein detoxification. Using CRISPR/Cas9, *akr1a1a-/-* zebrafish mutants with elevated endogenous ACR concentrations were created. A study conducted by Qi et al. found that *Akr1a1a* loss causes insulin resistance and impaired glucose homeostasis in larvae, as well as abnormal angiogenesis in the hyaloid vasculature and angiogenic retina vessels and GBM thickening in adults [25].

In T2DM patients, the carnosinase1 (*CNDP1*) gene has been identified as a factor that increases vulnerability to DN [65]. Carnosinase enzymes include serum carnosinase CN1 and cytosolic nonspecific carnosinase CN2 [66]. Carnosine lowered blood glucose levels, raised insulin, decreased albuminuria and vascular permeability, and repaired the glomerular ultrastructure in diabetic mice [67,68]. The pathological functions of the carnosine carnosinase system in zebrafish were investigated using a *CNDP1* knockout zebrafish model, which revealed that carnosine, anserine, and CN1 activity are available in zebrafish, with the ability to breakdown other dipeptides; furthermore, carnosine, *CNDP1-/-* animals are viable and grow normally, but have an altered amino acid profile. In addition, *CNDP1* deletion increases carnosine concentrations in vivo and protects against weight gain to some extent, but it is inadequate to prevent diabetes-related problems [26].

For flexible cell-type-specific expression, transgenic zebrafish lines expressing Grx1-roGFP2 in the cytoplasmic matrix and mitoGrx1-roGFP2 in mitochondria were produced under Gal4/UAS control (Tg(UAS: Grx1-roGFP2) and Tg(UAS:mitoGrx1-roGFP2)). The biosensors were preferentially produced in the pronephric tubules after crossing with the Gal4 line Tg(cdh17: Gal4), determined by the pronephros-specific cadherin 17 (cdh17) promoter [69]. Transgenic embryos were exposed to a reducing agent or an oxidant for 10 min, to examine the dynamic range and responsiveness of the Grx1-roGFP2 biosensor. The 405/488 nm excitation ratios of the biosensors in the cytosol and mitochondria decreased after 10 min of exposure to an uncoupling agent, carbonilcyanide p-triflouromethoxyphenylhydrazone, representing an uncoupling of oxidative phosphorylation which diminishes cellular oxidative stress. Uncoupling increases oxygen consumption in mitochondria, probably resulting in less oxygen radical production. Thus, the research conducted provides strong and adaptable techniques for studying redox dynamics in live animals [27].

*Dach1*, a transcription factor required for cell fate determination, is linked to the GFR. In research conducted where morpholino was injected, knock down of *Dach1* into fertilized eggs led to morphological abnormalities in the glomeruli of larvae, as well as down-regulation of nephrin and filtration barrier leakage. In contrast, to control biopsies, glomeruli from diabetic nephropathy patients revealed a substantial decrease in *Dach1* and synaptopodin, in comparison to control biopsies [28].

Podocytes express the *WHSC1L1-L* gene, and functional protein products have been found in these cells [70]. Although *WHSC1L1-L* has been found to bind nephrin, it has not been linked to any other podocyte-specific gene promoters, causing it to be inhibited or suppressed and, therefore, negating the stimulatory effects of WT1 and NF-B [71]. The injection of Whsc1l1 mRNA into zebrafish embryos resulted in lowering the expression of nephrin mRNA, but not podocin or CD2AP mRNA. In glomeruli, *WHSC1L1-L* and nephrin appeared during the S-shaped body stage. In glomerular podocytes, WHSC1L1 colocalizes with trimethylated H3K4 and reduces trimethylated H3K4 in the nephrin promoter regions. At the initial proteinuric stage of mouse nephrosis, nephron mRNA was upregulated in the glomerulus, which was linked to a decrease in WHSC1L1. In conclusion, these findings showed that *WHSC1L1-L* operates as a histone methyltransferase in podocytes and controls the expression of nephrin, which could help in maintaining the integrity of the glomerular filtration barrier’s slit diaphragm. *Nephrin* gene regulation is influenced by *WHSC1L1-L*. In addition to *WHSC1L1-L*, transcription factors and unknown molecules must modulate nephrin promoter activity in a complicated way [30].

PKA-mediated phosphorylation of HEXIM1 with serine-158 disturbs the dormant PTEFb/HEXIM1/7SK-snRNP complex through cAMP-PKA signaling. The cAMP pathway is important for the progression of autosomal dominant polycystic kidney disease (ADPKD), and P-TEFb has been found to be hyperactive in ADPKD kidneys in mice and humans. The zebrafish ADPKD model has also shown that genetic activation of PTEFb increases cyst development. Cystic kidneys, hydrocephalus, and dorsal curvature of zebrafish embryos are caused by MO-induced knockdown of *pkd2* [72]. Hexim1 and *pkd2* MO co-injection resulted in a more severe cystic renal phenotype than *pkd2* and control MO co-injection, and hexim1 knockdown enhanced cyst development in vivo. WT and S158E mutant human HEXIM1 mRNA were injected into the pkd2/hexim1 double mutant, to see if HEXIM1’s regulation of cyst formation was dependent on its P-TEFb sequestration function. Compared to EGFP mRNA injection, infusion of WT human HEXIM1 mRNA dramatically reduced cyst development, while injection of S158E mutant HEXIM1 mRNA failed to protect the cystic kidney phenotype [33].

There was a downregulation in the expression of *SLC34A1*, which is tubule specific in CKD cases. *sgRNAs* and CRISPR interference (CRISPRi) were used to corroborate the regulatory function of rs6420094 element on *SLC34A1* transcription by inactivating the rs6420094 element [73]. As a result of which, it was seen that this particular silencing of the element did reduce the expression of *SLC34A1*. The same gene, when knocked down in zebrafish, led to the formation of edema, which is an indicator of kidney dysfunction in zebrafish. Yet another element, rs2049805, was identified that also showed an association with renal functioning. It was also seen that genes such as GBAP1, THBS3, MTX1, and MUC1 were possible targets, because their promoters interplayed with rs2049805 located enhancer; however, GBAP1 was the nearest gene to the element [74]. Thrombospondin 3 (THBS3), a gene that is tubular-specifically expressed in humans, was experimented on in zebrafish by knocking out the gene, but there was no significant change in the phenotype [75]. Moreover, MTX1 gene knockout did show kidney injury. All of the above findings suggested that the cell type-specific epigenetic background might help identify cell type-specific functional variations and provide molecular annotation for non-coding SNPs [34].

#### 3.1.2. miRNAs

By controlling diabetic microangiopathy in diabetes-related diseases, such as retinopathy, nephropathy, wound healing, and cardiac damage, microRNAs and their target genes reveal novel processes and therapeutic targets. Emerging delivery platforms for modifying microRNA function or expression may be the next step in treating diabetes-related microvascular dysfunction and its clinical manifestations [76]. MiRNAs, which are roughly 22 nucleotides long, are noncoding short RNAs that have been discovered to be one of the most essential signal transduction mediators. MiRNAs bind to the target’s 3′UTR with altered complementarity in zebrafish, causing translational repression and inhibition [77].

The phenotypic and proteinuria development of zebrafish eggs injected with a *miR-143* mimic, or morpholinos specific to its targets syndecan and versican, were compared. In zebrafish eggs, overexpression of *miR-143* in one- to four-celled stages caused pericardial and yolk sac edema. Tg(l-fabp:DBP:EGFP) expressing 78 kDa green fluorescent plasma protein was utilized to further study the previously described zebrafish proteinuria model [78]. Plasma protein fluorescence was considerably reduced in *miR-143* injected zebrafish compared to controls, implying that this protein had been lost from circulation. At 48 h post-fertilization (hpf), the *miR-143* mimic was injected into the zebrafish’s cardinal vein, and no proteinuria or morphological defects were observed at 96 hpf. However, zebrafish injected with *miR-143* suffered pericardial effusions, yolk sac edema, and substantial loss of plasma proteins at 120 hpf. Syndecan-3, 4, and versican were all downregulated considerably. *MiR-143* overexpression or morpholino-mediated versican knockdown resulted in plasma protein loss, edema, podocyte effacement, and endothelial injury. Syndecan 3 and 4 knockdowns, on the other hand, did not affect the glomerular filtration barrier. For effective barrier function, versican and syndecan isoform expression is required. Podocyte-derived *miR-143* can affect the expression of glomerular glycocalyx proteins, which helps paracrine and autocrine cross-talk between podocytes and endothelial cells of glomerulus [29].

The extracellular matrix protein nephronectin (NPNT) is found in GBM. As the zebrafish anterior kidney has only one glomerulus and is structurally similar to humans, it can be used as a model for glomerular disease research [79]. The zebrafish glomerular filtration barrier is evolved at around 48hpf, according to fluorescent dextran injection tests [80]. The infusion of npntMO or a miR-378a mimic produces systemic edema in zebrafish larvae, and the identification of 70 kDa fluorescently tagged dextran injected into water is a microstructural cause of proteinuria and proteinuria, as well as the suggested blockage of podocytes with GBM thickening. VEGF-Aa, on the other hand, is a target for miR-378a-3p, which plays an essential role in glomerular barrier function, but being restricted to the endothelium [81]. The injection of recombinant zebrafish vegf-Aa protein was capable of recovering the phenotype generated by vegf-Aa-MO, although not the phenotype prompted by miR-378a-3p mimic injection, rejecting VEGF-Aa targeting by miR-378a-3p as a cause of the glomerular phenotype. Damage to podocytes and changes in the GBM can result from *NPNT* expression problems. As a result, miR-378a-3p might be a noninvasive marker of diagnosis or a therapeutic target for glomerular disorders [31].

*MiR-26a-5p* is upregulated in preeclamptic women’s placentas and plasma [82]. In hepatocellular carcinoma, *miR-26a-5p* has been shown to modulates *VEGF-A* expression [83]. *MiR-26a-5p* is mostly expressed by podocytes in the glomerulus [84]. The impact of *miR-26a-5p* on *VEGF-A* expression is mediated indirectly through the PIK3C2/Akt/HIF-1/VEGFA signaling pathway. Podocytes are the primary source of *VEGF-A*, *miR-26a-5p* regulates local *VEGF-A* expression. Overexpression of *miR-26a-5p* in zebrafish through microinjection of a particular *miR-26a-5p* mimic was sufficient to simulate preeclampsia glomerular alterations such as proteinuria, loss of function of glomerular endothelial cells, and podocyte foot process effacement [32].

## 4. Advantages of Using Zebrafish for Diabetic Nephropathy Studies

Zebrafish may generate hundreds of offspring every week and grow quite quickly. Zebrafish have a short reproductive cycle, making them ideal for drug testing on a large scale. In comparison with mice models, zebrafish is seen as more efficient and advantageous concerning the fast induction of diabetic phenotypes in the organism, which otherwise, in the case of mice models, would take months to induce. For the following reasons, zebrafish have developed unique traits that make them an ideal animal model for developmental and genetic studies: Female zebrafish can lay hundreds of eggs per week, the egg fertilization process is external and enables the production of haploid embryos, which are transparent, water-soluble drugs that can be quickly administered by adding them to water or through being injected, and the zebrafish genome has been fully sequenced.

Cell types, organs, and physiological systems are all preserved in these in vivo models and other vertebrate species. Furthermore, they have sufficient physiological complexity, as well as strong physiological and genetic similarities with humans. It is a completely sequenced genome and a genetically controllable creature; as well as their cost-effectiveness and high-throughput screening possibilities. Another advantage is the availability of diverse zebrafish strains, including over 1000 transgenic and mutant zebrafish strains.

In zebrafish, forward and reverse genetic techniques for gene-specific research are simple. Changed gene functions and symptoms may be explored using random mutagenesis in the embryo, to locate the responsible gene using a forward genetic method. Zebrafish screening is a viable method for identifying genes involved in vertebrate development. They have also been demonstrated to be effective for identifying candidate genes linked to human illness. In contrast, to reverse genetics, which involves disrupting genes of interest to assess their impact on the embryo, this is an impartial technique. Overall, employing zebrafish in metabolic diseases research has definite advantages. To better understand disease pathways, models for researching diabetes and its consequences and to provide new targets for disease treatment have been developed.

## 5. Limitations in Zebrafish Models

The use of zebrafish as a model for metabolic diseases faces more challenges than any other. Lifestyle, behavior, and socio-economic aspects all impact human metabolism, but zebrafish cannot replicate them. Individual zebrafish nutritional absorption is particularly difficult to track, and zebrafish development is influenced by numerous uncontrolled factors, such as temperature, body fat, and heredity; thus, it is not a reliable indicator of consumption. Metabolic procedures such as hormone testing and insulin resistance tests in zebrafish are substantially more challenging, because of the limits of recurring blood samples. Some operations, such as intraperitoneal injections, are difficult in zebrafish, but are simple in other animal models. Although controlling the concentration of a medication in water is simple, predicting how much drug will be absorbed by zebrafish is more complex. False-negative findings in toxicological screening, on the other hand, might be more concerning, since they can miss a compound’s considerable toxicity. Improved approaches for predicting or measuring drug ingestion might make zebrafish far more effective in drug research.

Another issue is obtaining enough zebrafish to perform large-scale, high-throughput screening. This is especially tough when dealing with transgenic strains that are delicate or inbred. As the zebrafish genome has a significant number of duplicate genes, destroying one gene copy is unlikely to damage the second copy, this characteristic precludes the development of knockout strains, making the translation of zebrafish to man one of the major challenges faced by the scientific community. However, programmable nucleases, such as transcriptional activator-like effector nucleases (TALENs), Clustered Regularly Interspaced Short Palindromic Repeats, and CRISPR-Associated Protein 9 can now overcome this constraint (CRISPR Cas9). These novel reverse genetics methods, as well as their ease of introduction and transplantation in zebrafish, will aid in a more thorough investigation of zebrafish system-level genetic function. Parental care is unclear, but it is critical in modeling some developmental diseases, such as autism, and may require other options. Certain brain areas (such as the cortex) are less developed in zebrafish than in mammals, and several CNS structures in zebrafish are still challenging to map to their human counterparts. Zebrafish have various drawbacks for the assessment of DN, including the fact that primordial renal cells in zebrafish are functionally significantly different from those in humans; and because zebrafish live in an aquatic environment, screening of some water-soluble medicines is another area of research.

## 6. Future Perspectives

In recent years, research into zebrafish models, human illnesses, and progressions has demonstrated the use of zebrafish in a variety of investigations. The distinctive peculiarity of zebrafish, which allows for quick screening, and the differentiation of molecular alterations depending on gender and organ specifications, makes them an excellent candidate for research. Zebrafish may also be utilized for unbiased, high-throughput chemical screening, which increases their use in in vivo drug development. The zebrafish has introduced itself as a potential option for drug discovery, by providing access to a wide range of genetic tools, a remarkable similarity to humans and our genetic mutations, rapid evolution, and limited ethical issues.

Zebrafish are increasingly being used in precision medical research, particularly in endocrine problems. Several zebrafish cancer research studies have been carried out, in order to create various cloned xenograft models for target validation, medication safety, toxicity, and effectiveness investigations, among other things. Since zebrafish have comparable benefits in cancer studies, their potential for tailored regenerative therapy offers promise. Furthermore, advancements in zebrafish model visualization methods, gene targeting, and chemoprevention may increase future prospects and insights. The translational utility of zebrafish as a standard platform for drug screening and testing is ultimately determined by their renal physiology. Zebrafish model systems may now be used for preclinical and customized medical screening for human illnesses, owing to advancements in CRISPR/Cas9 technology and applications.

## Figures and Tables

**Figure 1 vetsci-09-00312-f001:**
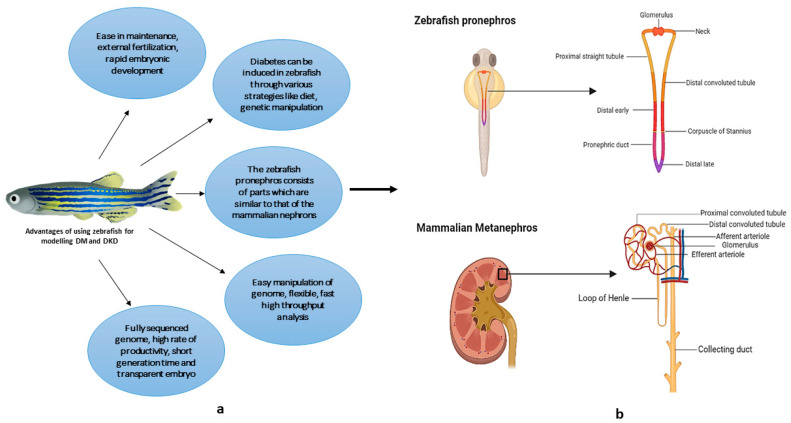
(**a**) Representation of the advantages of the zebrafish as a DM and DKD model; (**b**) similar segmentation pattern shared by Zebrafish pronephric and Human Metanephric nephrons (Created using Biorender.com, accessed on 20 February 2022).

**Figure 2 vetsci-09-00312-f002:**
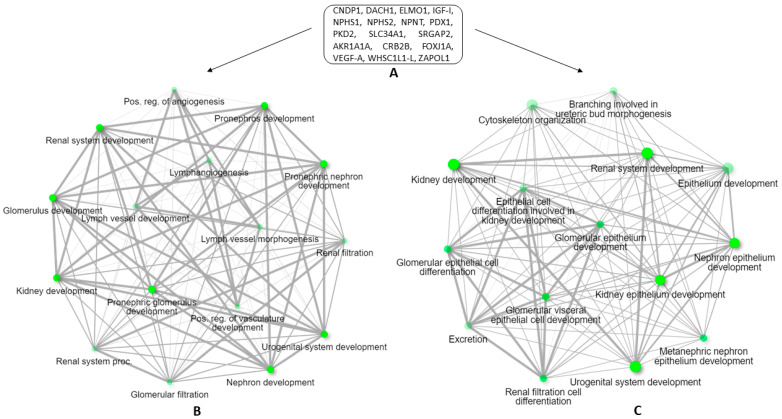
Significantly enriched gene ontology of biological processes corresponding to the DN associated genes in zebrafish. List of genes (Table 1) studied in DN using zebrafish model (**A**); the figure further depicts the biological processes these genes are involved in in zebrafish (**B**), human (**C**). Darker nodes (dot) are more significantly enriched gene sets. Bigger nodes represent larger gene sets. Thicker edges (line) represent more overlapped genes. For the visualization, we used graphical gene-set enrichment tool ShinyGO [17].

**Table 1 vetsci-09-00312-t001:** Selected genes and experimental models of diabetic nephropathy in zebrafish.

Gene	Stage	Effects	Reference
*IGF-I receptor*, *U6x*:*sgRNA*	Larvae	CRISPR/Cas9 knockdown of liver insulin receptor causes postprandial hyperglycemia and fasting hypoglycemia. Dominant-negative expression of *IGF-I receptor* (IGFIR) in skeletal muscle causes an elevation in fasting blood glucose.	[18]
*Nephrin* and *Podosine*	Embryo	Morpholino-mediated knockdown of *nephrin* and *podosine* caused the disappearance of the podocyte process.	[19]
*Crb2b*	Embryo	Morpholino-mediated knockdown of the *Crb2b* protein caused an overall disruption of the podocyte process, including loss of the slit diaphragm.	[19]
*foxj1a*	Embryo	Induction of *foxj1a* increased the length of cilia and boosted cilia beat in response to kidney damage.	[20]
*ELMO1*/*DOCK180*	Embryos	ELM O1 knockout exhibits alterations in the zebrafish pronephric structure.	[21]
*pdx1* *pdx1sa280*	Adult	Mutations lead to reduced body size and reduced survival. Beta-cell numbers and insulin levels are decreased and the exocrine pancreas is defined, but acinar differentiation is impeded. Overall pancreatic islet size was significantly reduced in the mutants.	[22]
*zApoL1*	Larvae	When the *zApoL1* gene was knocked off, the larvae suffered significant pericardial edema. With knockdown of the expression of *zApoL1* gene, the larvae developed having critical pericardial edema.	[23]
*SRGAP2a*	Larvae	*SRGAP2a* knockdown causes podocyte foot process effacement, a disrupted slit diaphragm, and a disorganized glomerular filtration barrier, as well as reduced GFP expression.	[24]
*Akr1a1a*	Larvae	Knockout induced impaired glucose homeostasis, followed by abnormal angiogenesis in the hyaloid vasculature of the larvae, resulting in angiogenic retinal vessels and GBM thickening in adults.	[25]
*CNDP1*	Embryos	Knockouts can increase carnosine levels in vivo and prevent some weight gain to an extent, but not enough to prevent the complications caused by diabetes.	[26]
*ssGrx1-roGFP2*	Embryos	When crossed with the Gal4 strain, propelled by the anterior kidney-specific cadherin 17 (cdh17) promoter, biosensors were precisely expressed in the pronephric tubules.	[27]
*DACH1*	Larva	Knockdown of zebrafish ortholog Dachd1 induces glomerular morphological changes with downregulation of *nephrin* and leakage of the filtration barrier.	[28]
*miR-143*	Eggs	Overexpression of *miR-143* in zebrafish eggs at the one- to four-cell stages caused a phenotype, resulting in pericardial and yolk sack edema.	[29]
*WHSC1L1-L*	Embryos	Injection of Whsc111 mRNA into the embryo showed a clear reduction in *nephrin* mRNA, but not podocin and CD2AP mRNA.	[30]
*NPNT*	Larvae	Knockdown resulted in generalized edema, and uncovering of injected 70 kDa fluorescence-labeled dextran in the water showed proteinuria and podocyte effacement, and thickening of the GBM	[31]
*VEGF-A*	Larvae	The glomerular phenotype was salvaged by injecting recombinant zebrafish vegf-Aa protein.	[31]
*miR-26a-5p*	Embryo	Proteinuria, endothelial cell enlargement, edema, damage of glomerular endothelial fenestration, and podocyte foot process effacement are all symptoms of overexpression in preeclampsia.	[32]
*pkd2*	Embryos	Knockdown resulted in cystic kidney, hydrocephalus, and dorsal axis curvature in zebrafish embryos.	[33]
*SLC34A1*	Larvae	In zebrafish, *SLC34A1* knockout caused edema and impaired kidney function.	[34]

## Data Availability

Not applicable.

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
