# Peer review of "Zebrafish: A Model to Study and Understand the Diabetic Nephropathy and Other Microvascular Complications of Type 2 Diabetes Mellitus"

_vetsci, 2022, doi:10.3390/vetsci9070312_

Round 1

Reviewer 1 Report

Comments to the authors:

The authors have submitted a timely, interesting and comprehensive review. It is well structured, reads easily and summarizes the most important advancements made in zebrafish studying diabetes and its complications. I have a few suggestions to improve the manuscript.

1.       The title is misleading, because the review focuses on diabetic nephropathy only and not on all microvascular complications including diabetic neuropathy and retinopathy as well. I suggest to modify the title accordingly.

2.       The abstract could be more focused on major advancements in the field.

3.       Line 136: the STZ model in zebrafish does not work well and the original protocol published by the Intine group could not be reproduced later.

4.       Chapter 3 is pretty long and could be better structured to enhance comprehension (as it was done for chapter 2).

5.       Line 209; the PDX1 mutant were recently introduced by the Kroll group as a model for DN.

6.        Chapter 4: In mice, it takes months to induce DN, in fish only a few days. I would consider the fast induction of diabetic phenotypes in zebrafish as one of the most important advantages.

7.       Chapter 6: Chemical screens and drug discovery are definitely essential, but I think the translation from zebrafish to man is the major challenge for the scientific community.

Author Response

Comments to the authors:

The authors have submitted a timely, interesting and comprehensive review. It is well structured, reads easily and summarizes the most important advancements made in zebrafish studying diabetes and its complications. I have a few suggestions to improve the manuscript.

Comment 1: The title is misleading, because the review focuses on diabetic nephropathy only and not on all microvascular complications including diabetic neuropathy and retinopathy as well. I suggest to modify the title accordingly.

Response: Thank you for your suggestion. Taking your valid comment into consideration, we have changed the title to a more specific one (Refer Page No. 1).

Comment 2:  The abstract could be more focused on major advancements in the field.

Response: Thank you for making this valid point. We have added We have added the suggested content to the manuscript (Refer: Abstract, page no.1; line 33-36).

Comment 3:  Line 136: the STZ model in zebrafish does not work well and the original protocol published by the Intine group could not be reproduced later.

Response: Thank you for pointing out this, we have made the changes (Refer: 2. Studying diabetes using zebrafish models, Page No. 4; line 144-147).

Comment 4: Chapter 3 is pretty long and could be better structured to enhance comprehension (as it was done for chapter 2).

Response: Thank you for the suggestion. We have incorporated the necessary modifications (Refer: 3.1. Molecular level approaches to model DN in zebrafish, 3.1.1. Genes, 3.1.2 miRNAs; Page Nos. 5-9).

Comment 5:  Line 209; the PDX1 mutant were recently introduced by the Kroll group as a model for DN.

Response: Thank you for this suggestion. We have added the suggested content to the manuscript (Refer3.1. Molecular level approaches to model DN in zebrafish, 3.1.1. Genes, Page No. 5; line 225-228).

Comment 6: Chapter 4: In mice, it takes months to induce DN, in fish only a few days. I would consider the fast induction of diabetic phenotypes in zebrafish as one of the most important advantages.

Response: Thank you for your valuable comment, we have incorporated your suggestion (Refer: 4. Advantages of using zebrafish for diabetic nephropathy studies, page No. 14; line 593-595).

Comment 7:  Chapter 6: Chemical screens and drug discovery are definitely essential, but I think the translation from zebrafish to man is the major challenge for the scientific community.

Response: Thank you sharing your valuable thought, we have included this point the under the limitations section (Refer: 5. Limitations in Zebrafish Models, Page No.15; line 636-637).

Reviewer 2 Report

In this review, author concerned that Diabetes Mellitus (DM) is a complex metabolic disease that has had a global impact and imposed an unsustainable burden on healthcare systems in both developed and developing countries. The Zebrafish has established itself as a disease model organism. The organism can also be used in research and connected genetic defects to the emergence of diabetic complications. When combined with genetic modifications and/or mutant or transgenic fish lines, the zebrafish has become one of the most valuable models for researching DN. Overall this is a comprehensive review, but at some point may need further demonstration.

1. the title mentioned "Diabetes Mellitus", but author only discussed T2DM in the paper. Although T2DM is a very important subtype of diabetes, but there certainly also some valuable models for T1DM, I would suggest author include model for T1DM as well, or specify that this paper is T2DM only in the title.

2. The title mentioned "Microvascular Complication", but in the content, og subsection about Microvascular Complication is nephropathy. Meanwhile, Retinopathy is another complication that frequently seen in diabetic patients. Similar to suggestion 1, I would suggest author include model for retinopathy as well, or specify that this paper is nephropathy only in the title.

3. It would be nice if author could add 1 or 2 figure to further elucidate the mechanism of these genes involved in diabetic Microvascular Complication.

Author Response

In this review, author concerned that Diabetes Mellitus (DM) is a complex metabolic disease that has had a global impact and imposed an unsustainable burden on healthcare systems in both developed and developing countries. The Zebrafish has established itself as a disease model organism. The organism can also be used in research and connected genetic defects to the emergence of diabetic complications. When combined with genetic modifications and/or mutant or transgenic fish lines, the zebrafish has become one of the most valuable models for researching DN. Overall this is a comprehensive review, but at some point may need further demonstration.

Comment 1: the title mentioned "Diabetes Mellitus", but author only discussed T2DM in the paper. Although T2DM is a very important subtype of diabetes, but there certainly also some valuable models for T1DM, I would suggest author include model for T1DM as well, or specify that this paper is T2DM only in the title.

Response: Thank you for your valuable comment, we have changed the title of the paper to a more specific one (Refer Page No. 1).

Comment 2: The title mentioned "Microvascular Complication", but in the content, og subsection about Microvascular Complication is nephropathy. Meanwhile, Retinopathy is another complication that frequently seen in diabetic patients. Similar to suggestion 1, I would suggest author include model for retinopathy as well, or specify that this paper is nephropathy only in the title.

Response: Thank you for your suggestion, we have changed the title of the paper to a more specific one (Refer Page No. 1).

Comment 3: It would be nice if author could add 1 or 2 figure to further elucidate the mechanism of these genes involved in diabetic Microvascular Complication.

Response: Thank you for valuable suggestion. We have incorporated an additional figure (Figure 2: Significantly enriched Gene Ontology of Biological Process corresponding to the DN associated genes in zebrafish) (Refer: Page No.14).

Reviewer 3 Report

The topic of paper of  Charles Scarchil et al.   is very interesting and addresses a model for interesting future analyses, especially for the clinic. I have no particular comments on it.

Author Response

The topic of paper of  Charles Scarchil et al.   is very interesting and addresses a model for interesting future analyses, especially for the clinic. I have no particular comments on it.

Response: I am thankful to the reviewers for providing me a positive comment on the manuscript. Thank you once again.